# Gendered lives, gendered Vulnerabilities: An intersectional gender analysis of exposure to and treatment of schistosomiasis in Pakwach district, Uganda

**Sarah N. Ssali**[1]*, **Rosemary Morgan**[2], **Salama Nakiranda**[1], **Christopher K. Opio**[3], **Mariam Otmani del Barrio**[4]

**1** Makerere University School of Women and Gender Studies, Kampala, Uganda, **2** Department of International Health, John Hopkins Bloomberg School of Public Health, Baltimore, Maryland, United States of America, **3** Aga Khan University Hospital, Nairobi, Kenya, **4** UNICEF/UNDP/World Bank/WHO Special Programme for Research and Training in Tropical Diseases (TDR), World Health Organization, Geneva, Switzerland

* sarah.ssali@mak.ac.ug

## Abstract

### Introduction

Schistosomiasis is a neglected tropical disease (NTD) that is endemic in Uganda, despite several interventions to eliminate it. It is transmitted when people infected with it pass on their waste matter into fresh water bodies used by others, consequently infecting them. Several studies have demonstrated gender and age differences in prevalence of schistosomiasis and NTDs such as lymphatic filariasis and soil transmitted helminths. However, few intersectional gender analysis studies of schistosomiasis have been undertaken. Using the World Health Organisation (WHO)'s intersectional gender analysis toolkit, this study was undertaken to identify which social stratifiers most intersected with gender to influence vulnerability to and access to treatment for schistosomiasis disease, to understand how best to implement interventions against it.

### Methodology

This was a qualitative study comprising eight focus group discussions (FGDs) of community members, disaggregated by age, sex and location, and 10 key informant interviews with health care providers and community leaders. The Key informants were selected purposively while the community members were selected using stratified random sampling (to cater for age, sex and location). The data was analysed manually to identity key themes around gender, guided by a gender and intersectionality lens.

### Results

The study established that while the River Nile provided livelihoods it also exposed the community to schistosomiasis infection. Gender relations played a significant role in exposure to and access to treatment for schistosomiasis. Traditional gender roles determined the

**Data Availability Statement:** All data underlying the findings described in their manuscript has been

uploaded as S1 Data and will be made freely available to other researchers.

**Funding:** This investigation received financial support from the UNICEF/UNDP/World Bank/WHO Special Programme for Research and Training in Tropical Diseases (TDR - www.who.int/tdr) grant number B80216 to SS, RM, SN and CO who participated in study design, data collection and analysis, decision to publish, and preparation of the manuscript. MOdB is a staff of the UNICEF/UNDP/World Bank/WHO Special Programme for Research and Training in Tropical Diseases (TDR) at the World Health Organization. TDR staff participated in the conceptualization and study design, subsequent manuscript reviews and revisions. The authors alone are responsible for the views expressed in this article and they do not necessarily represent the views, decisions or policies of the institutions with which they are affiliated.

**Competing interests:** The authors have declared that no competing interests exist. The authors alone are responsible for the views expressed in this article and they do not necessarily represent the views, decisions or policies of the institutions with which they are affiliated.

activities men and women performed in the private and public spheres, which in turn determined their exposure to schistosomiasis and treatment seeking behaviour. Gender relations also affected access to treatment and decision making over family health care. Men and some women who worked outside the home were reported to prioritise their income earning activities over seeking health care, while women who visited the health facilities more regularly for antenatal care and to take sick children were reported to have higher chance of being tested and treated in time, although this was undermined by the irregular and infrequent provision of praziquantel (PZQ) mass drug administration. These gender relations were further compounded by underdevelopment and limited economic opportunities, insufficient health care services, as well as the respondent's age and location.

## Conclusions

The study concludes that vulnerability to schistosomiasis disease and treatment occurred within a complex web of gender relations, culture, poverty, limited economic opportunities and insufficient health services delivery, which together undermined efforts to eliminate schistosomiasis. This study recommends the following: a) increased public health campaigns around schistosomiasis prevention and treatment; b) more regular PZQ MDA at home and schools; c) improved health services delivery and integration of services to include vector control; d) prioritising NTDs; e) providing alternative economic activities; and f) addressing negative gender norms that promote social behaviours which negatively influence vulnerability, treatment seeking and decision making for health.

### Author summary

Schistosomiasis is a waterborne NTD endemic to Uganda and many other countries in Africa, Asia and Latin America. Schistosomiasis is one of the prominent NTDs in Uganda, the others being trachoma, onchocerciasis, buruli ulcer, soil-transmitted helminths, and elephantiasis. In Uganda, schistosomiasis only ranks behind malaria among parasitic diseases as a cause of poor health. Schistosomiasis has several socio-economic drivers, which include poverty, limited economic opportunities, poor service delivery and negative socio-cultural and gender norms. Yet the conventional responses to it have often been biomedical, using PZQ MDA. This study applied intersectional gender analysis to study how gender relations intersected with other social stratifiers to shape women's and men's experience of exposure to and treatment of schistosomiasis. It established that traditional gender roles determined the activities men and women performed at home and in public, which in turn determined their exposure to schistosomiasis. Gender roles were also reported in the fishing industry, where men predominantly fished while women processed and sold fish. While men were considered to be the epicentre of infection, women were not any safer since their traditional gender roles of cooking and washing brought them in direct contact with infected water multiple times daily. Gender roles also determined access to prompt health care with men (and some women who worked) prioritising making money instead of treatment, while women who were charged with taking children to hospital had opportunity to be tested and treated in time. Apart from gender differences were other factors such as poverty and limited health care services which compounded gender inequality. For example, the insufficient supply of PZQ undermined women's

opportunities to access prompt health care, while long waiting time deterred men and working women from seeking health care. Meanwhile with poverty the community had no alternative means of survival except the River Nile, despite its being infested with schistosomiasis bearing vectors. The study concludes that gender has a significant role to play in understanding and responding to schistosomiasis and makes the several recommendations including: mainstreaming gender in schistosomiasis control strategies; prioritising NTDs elimination in health systems; less vertical programmes in preference of integrated health care services, including in schistosomiasis control; poverty eradication; and more research in intersectional gender analysis in NTDs.

## Introduction

Schistosomiasis is an acute and chronic parasitic NTD caused by blood flukes (trematode worms) of the genus *Schistosoma* [1], [2]. It is a water-borne parasitic and highly endemic NTD [3]. NTDs refer to a set of 13 tropical diseases largely afflicting the world's poor who have limited access to water and sanitation services, but which have not received much attention [4]. Of the 13 NTDs, six have been identified for elimination, including Schistosomiasis, Dracunculiasis (Guinea Worm Disease), Lymphatic Filariasis, Onchocerciasis, Soil-transmitted Helminths (STH) and Trachoma [5]. NTDs are endemic in several tropical countries in Africa, Asia and Latin America. Schistosomiaisis transmission occurs when people suffering from schistosomiasis contaminate freshwater sources with their excreta containing parasite eggs, which hatch in water. This is the main reason why water bodies, with freshwater snails that serve as a host for development of free-living larvae, remain the main source of infection [3]. Human beings of all ages get infected upon contact with freshwater bodies harbouring free living larvae of the parasite that causes schistosomiasis [3].

Schistosomiasis is one of the prominent NTDs affecting the country, the others being trachoma, onchocerciasis, buruli ulcer, soil-transmitted helminths, and elephantiasis [6]. In Uganda, schistosomiasis ranks behind malaria among parasitic diseases as a cause of poor health [7]. For example the first national representative survey conducted in 2016–2017 established that the prevalence of schistosomiasis was 25.6%, with 2–4 years olds having the highest prevalence at 36.1% [3]. Fifty-five percent of the population is considered at risk [8], [6], with the most affected being fisher folks [9] and lately those working in rice paddy fields [10]. Though prominent along Lake Albert and Lake Victoria shores, schistosomiasis has been found at high altitudes in Mount Elgon's crater lakes and Tooro [8,11].

The most common strategy to control schistosomiasis is by PZQ mass drug administration (MDA) [12], undertaken by targeting school age children (SAC) for morbidity control and elimination [13]. But due to the inadequate supply of PZQ in Africa due to the high disease burden [13], other community strategies have been undertaken such as improved water and sanitation, community education [14], deploying community leaders in drug administration, targeting pregnant women and children, and pre-treatment snacks as a strategy to increase uptake of PZQ [15]. Being diseases of poverty, NTDs are associated with social problems such as social stigma and gender relations, which decrease people's uptake of health information and or care [16,17]. Yet, most of programmes targeting them are vertical programmes such as the national Ministries of Health (MoH) Schistosomiaisis Control Programmes, which neither mainstream social issues such as gender mainstreaming [18], nor develop synergies among the several other actors providing health care in the same area, undermining the efficiency and effectiveness of programme implementation.

Most research on schistosomiasis has been gender neutral, only studying women as a statistical category [18]. Few studies have focused on how gender relations in schistosomiasis endemic areas intersect with other social stratifiers to affect vulnerability to and treatment of the disease. For example, research among juveniles showed no specific gender disparities in disease. While Kabateraine et al. [19] found women to be less heavily infected than males, the structure drivers for the disparity were not explored. Others have simply targeted pregnant women and juveniles for prevention and control [20]. Therefore, this study used the Intersectional Gender Analysis Toolkit [21] to explore how gender relations and other social stratifiers intersect to shape vulnerability to, and access to treatment for schistosomiasis disease by men and women, in order to understand how best to implement interventions against schistosomiasis.

Gender refers to the socially constructed roles, behaviours, activities, attributes and opportunities that any society considers appropriate for men and women, boys and girls and people with non-binary identities [21]. Gender is often relational, shaping how men/boys, women/girls and people with non-binary identities interact with each other and the world around them. Due to its social construction, gender frequently varies through spaces, contexts and time, as individuals construct differing roles and identities that are shaped by broader political, social and economic circumstance [21]. Gender as a power relation shapes vulnerability or risk of disease, access to and utilization of health services, as well as the decision making relating to health at household level [17]. However, gender is just one axis of social advantage/disadvantage within society, the others being age, ethnicity, religion, location, and policy to mention a few.

Intersectional gender analysis is the process of analysing how gender power relations intersect with other social stratifiers to affect people's lives; create differences in needs and experiences; and how policies, services and programmes can help to address these differences [21]. It is concerned with how people are positioned in different structures of power such as race, class and ethnicity and how these structures of power intersect [22]. It arises from the understanding that other structures of power such as race and class modify gender relations, shaping the difference in experiences of different men and different women. NTDs such as schistosomiasis occur in contexts of economic and social marginalization. Using an intersectional gender lens was key to highlighting how men and women in the predominantly rural Pakwach District, with limited opportunities, experienced vulnerability to schistosomiasis and constraints to health and health care access.

## Methods

### Ethics statement

Ethical clearance for this study was secured at three levels including, Vector Control Division Ethics Committee (Ministry of Health, Uganda Permit No. VCDREC/118), Uganda National Council of Science and Technology (Uganda, Permit No. SS 5157) and the WHO Ethics Review Committee. In addition, Pakwach District Office provided administrative clearance for the study during fieldwork. In undertaking this study, all participants provided written informed consent. Furthermore, their identities in this paper are anonymised.

### The study setting and population

Pakwach District is located in the West Nile region of Uganda. It is a new district, carved out of Nebbi district. It borders the Democratic Republic of the Congo to the Southwest, Arua District to the North West, Nebbi District to the West, Buliisa to the South, and Nwoya and Amuru districts to the East. The main communities in Pakwach are the Jonam and the Alur

tribes though the Alur are the most prominent ethnic group. The River Nile goes through Pakwach District, providing livelihoods for the mostly fishing communities we interviewed.

Pakwach district was selected because schistosomiasis was endemic, despite several attempts to eliminate it such as solar water sterilization, community drives to boil drinking water and construct pit latrines, and PZQ MDA for prevention and treatment of schistosomiasis. Safe water coverage in the district ranged from 17% in the urban town council to 95% in rural Alwii sub county [23]. However, 80 water points had not been functional for over five years and were considered abandoned [23]. The solar water sterilization kit was supposed to reduce the frequent need for fuel wood but its prohibitive cost of between 150,000UGX to 200,000UGX limited its usage. Moreover, by the time of this study, this project had long ended.

PZQ MDA was carried out through the MoH Schistosomiasis Control Programme. Drugs are donated by WHO and Merck and distributed by MoH through the district by health care workers at Pakwach health Centre IV to schools, some households, and at landing sites. Priority was given to school going children from schools selected by the District and the MoH. Although PZQ was an essential drug, its supply was infrequent. Nearly all supplies of PZQ for MDA were donations.

We collected data from an urban and peri-urban area within Pakwach district, to explore how differences in gender and location affected men's and women's vulnerability to disease and its treatment. The two areas selected were both close to water bodies. Pakwach town council, located on the banks of the River Nile was selected to represent the urban, while Panyimur sub-County, located on the banks of the River Nile but closer to Lake Albert was selected to represent the rural. Pakwach town council was the main "town", where the district headquarters are located. Panyimur, which is nearest to Lake Albert, was where the Panyimur Landing Site was found. The study population had limited education and economic opportunities, with most relying on fishing, snail mining and farming for survival.

## The study participants

Using an intersectional gender lens, study participants were purposively selected, paying attention to gender, occupation, age and location. There were two categories of participants: community members and key informants. 40 community members drawn from the two localities of Panyimur and Pakwach town council were disaggregated by gender, age categories and location and subjected to focus group discussions (FGDs). Specifically, the eight FGDs, comprising two groups of women aged 18 to 45 years, two groups of men aged 18 to 45 years, two groups of women aged 46 to 65 years, and two groups of men aged 46 to 65 years as illustrated in Table 1 below:

The key informants comprised six health workers and four community leaders. The six health workers included four males and two females, purposively selected because they had dealt with schistosomiasis cases for more than one year. The selected health workers were further endorsed by the District Health Officer, who had been involved in preventing and controlling schistosomiasis for more than 10 years. The four community leaders were purposively selected for their role in schistosomiasis control activities. All the participants had been engaged in community prevention through methods such as MDA.

## Data collection methods

This study used qualitative data collection methods, namely key informant interviews and FGDs, preferred for their attention to nuance and detailed discussion of experiences. 10 key informant interviews were held with the six health workers and four community leaders.

**Table 1. Illustrating FGDs distribution.**

| FGD | Unit | Number of Participants |
|---|---|---|
| Pakwach FGD Male (18–45) | 1 | 5 |
| Pakwach FGD Male (46–60) | 1 | 5 |
| Pakwach FGD Female (18–45) | 1 | 5 |
| Pakwach FGD Female (46–60) | 1 | 5 |
| Panyimur FGD Male (18–45) | 1 | 5 |
| Panyimur FGD Male (46–60) | 1 | 5 |
| Panyimur FGD Female (18–45) | 1 | 5 |
| Panyimur FGD Female (46–60) | 1 | 5 |
| **Total** | **8** | **40** |

Source: Fieldwork

Given the COVID19 situation, key informant interviews were mainly conducted on phone and lasted not more than an hour. FGDs were conducted in the community of study, to establish the commonly held views about gender relations and schistosomiasis vulnerability and treatment. Each FGD comprised five participants and lasted between 45 minutes to one hour. The responses of the different office bearers and community members, individually and within a group, were key to identifying the power relations arising from gender, class and age.

FGDs were conducted in the community of study, to establish the commonly held views about gender relations and schistosomiasis vulnerability and treatment. Each FGD comprised five participants and lasted between 45 minutes to one hour. The responses of the different office bearers and community members, individually and within a group, were key to identifying the power relations arising from gender, class and age.

FGDs were conducted in the community of study, to establish the commonly held views about gender relations and schistosomiasis vulnerability and treatment. Each FGD comprised five participants and lasted between 45 minutes to one hour. The responses of the different office bearers and community members, individually and within a group, were key to identifying the power relations arising from gender, class and age. The study questions were guided by the Gender relations framework that emphasizes gender as a power relation and driver of inequality [21]. Accordingly, this gender framework organizes gender power relations into four categories: who has what (access to resources); who does what (the division of labour, roles and everyday practices); how values are defined (social norms, ideologies, beliefs and perceptions) and who decides (rules and decision-making), and interrelationship between them, as well as highlight ways in which power is negotiated and changed at the individual and structural level. Guided by this framework, the questions for this study focused on: i) the roles performed by men and women and how these affected their exposure to infection; ii) the decision-making processes and pathways within the household and how these affected access to treatment; iii) contemporary lifestyle and their ability to change, to assess the norms and values they had about the activities they engaged for a living, and the possibilities of change.

## Data analysis

The data for this study was triangulated and analysed manually. FGD data, which had been conducted in Alur, were translated into English and transcribed according to the different sex, age and area categories. Key informants' interviews which had been conducted in English were transcribed and categorized according to the sex and position of the key informant. Each interview was then manually coded according to a coding frame developed for the study.

Themes were developed to identify key axes of power and how they interacted with gender relations across different categories.

## Results

This section presents the findings from the study under different emerging themes. The findings emphasise gender roles and exposure to infection, gender differences in men's and women's access to treatment, and changes required at community and health system level to curb exposure to infection. All these findings were collected and are presented in a gender disaggregated manner. Given that the study was largely qualitative, quotes from different categories of respondents are presented to illustrate the nuances in their views to infection and treatment.

### a) Gender and perception of vulnerability to schistosomiasis

The respondents were asked what predisposed men and women to infection. Generally, all were exposed from the activities that brought them into contact with the water bodies. Specifically, the main activities that predisposed them to infection were listed as: fishing, moving in contaminated water, snail mining, defecating and urinating in water and in the bushes, drinking unboiled water, fetching water, swimming, cleaning fish from water, farming (rice and tomato) on river banks, collecting river sand, brushing teeth using river sand, bathing in the river, washing clothes, drinking contaminated water, eating half-cooked fish, and constructing pit latrines along the River Nile, which contaminates the water.

A disaggregation by the gender and age category of the FGD showed most FGD participants considered men to be more at risk save for the FGDs of Pakwach females (old and young) and Panyimur young males as illustrated in Table 2 below:

The FGDs who perceived men to be most at risk considered the time men spent fishing compared to that women spent getting into contact with water for domestic use:

> "When you consider the duration men take using the lake compared to the time women take to buy fish and clean it then [take it] back home, there is a very big difference. The only way women are at risk of getting bilharzia is fetching water because they do it daily, and here in Panyimur some women are always at the garden digging, and if you say they do get the disease from the farms, there I do agree it can happen. But sincerely speaking, it's the men who go fishing and process the fish at most times. For instance, a person who goes fishing for about one month in the lake [man] and the one who goes in for thirty minutes or two hours in the Lake [woman] are not the same. (Panyimur Male FGD 46–65 years)

**Table 2. Illustrating Perceptions of Most at Risk.**

| FGD | Category | |
|---|---|---|
|  | **Men** | **Women** |
| Pakwach FGD Male (18–45) | *** |  |
| Pakwach FGD Male (46–60) | *** |  |
| Pakwach FGD Female (18–45) |  | *** |
| Pakwach FGD Female (46–60) |  | *** |
| Panyimur FGD Male (18–45) |  | *** |
| Panyimur FGD Male (46–60) | *** |  |
| Panyimur FGD Female (18–45) | *** |  |
| Panyimur FGD Female (46–60) | *** |  |

Source: Fieldwork

Furthermore, men were perceived to be at risk because of activities such as swimming, getting drunk and falling in the water, bathing in the river or drinking water directly from the river as illustrated below:

". . . Also swimming in the water, men like swimming in the water. Since the activity is done in the lake this exposes them more than women." (Panyimur Female FGD, 46–65 years)

". . . we men like bathing in the river for more time than the women. Men like bathing direct from the source. To add on to that, us men normally when thirsty drink this water directly from the river whenever we are fishing and this increases the risk of getting the disease than women. They [men] use the very water they are in. . . .". (Pakwach FGD Male, 46–65 years)

"I would like to add about these men who go to relax and end up getting drunk, fall in the swampy areas rolling in the mud and end up getting the disease. It is the drinking of alcohol." (Pakwach FGD Female, 46–65 years)

These quotes refer to men getting in contact directly with infected water from leisure activities, not directly related to fishing to earn a living. That women were not mentioned in relation to these activities could point to women's limited leisure time, either due to culture or time constraints. Those who considered women to be more at risk did so because of women's gender roles, which exposed them from the several times they came into contact with the water body fetching water for domestic use:

"A woman can go to the river to fetch water five (5) times, and a man may go fishing once a week. This fetching of water alone puts this woman at a high risk of getting infected than this man who goes fishing once a week. They like accessing water for drinking, yeah for domestic use. In our culture, it's the women to do that. It is their role. That is why they are still following that cultural belief of women fetching for the home, and it is very important. When piped water was brought, there was lifestyle changes, but still women preferred to get water from the river. That is why they can move five (5) to six (6) times fetching this water just like what Master had said". (Pakwach Male FGD 18–45 years)

The foregoing emphasizes coming into contact with the water bodies as the main source of exposure to infection. However, they highlight its gendered nature, owing to the different activities of men and women, which structured how women and men interacted with the water bodies as illustrated below.

## b) Gender roles and exposure to schistosomiasis infection

Gender roles were reported to influence men's and women's vulnerability to infection. Gender roles refer to responsibilities associated with our biological set-up or the expected duties and responsibilities, rights, and privileges of men and women/boys and girls that are dictated by cultural factors. These roles are shaped by society: influenced by religion, economy, cultural attitudes, and the political system. They are learnt through the process of socialisation and vary from one culture to another. Gender roles can be classified as productive, reproductive or community managing roles [24]. Depending on one's class, access to resources and positionality, different gender undertake different roles, which predispose them differently to disease and shape their experience to disease. For example, in the research setting, it was men were expected to head the households, which included providing for the family and deciding on health care, while women were expected to care for the family, which included taking sick

children to hospital. These gender roles predisposed men and women differently to infection by determining the economic activities they engaged in, as well as the gender division of labour within the fishing industry.

> "... when we look at men being actually at the epicenter of all these I mean, I can say that we are the breadwinners at home. You as a man, you are married, you wake up very early in the morning and the woman asks for food, you must go and fish. That is one thing. So, we look for sources of living and how to earn a living in our homes. That is why I say that we are at the epicenter of all these. At the peak of getting infected". (Panyimur Male FGD, 18–45 years)

> "...Women do get it [schistosomiasis] from fetching of water as they go and stand in the water for some time as they fetch and also washing of clothes. Secondly, this water fetched by women are never left under sunshine or boiled. They use it direct for bathing children, cooking and drinking, which leads to infection. (Panyimur Female FGD 18–45)

From the above, gender roles put men at the 'epicenter' of infection, without sparing women from infection. Women's and children's domestic chores equally exposed them. Direct utilization of the water by women could have arisen from either not knowing the dangers of doing so, or from limited access to fuel for boiling the water.

A gender division of labour existed within the fishing industry. For example, actual fishing was a role of men, which predisposed them to infection due to the long periods they spent in the water and the habits they exhibited when fishing such as swimming naked in the water, standing in dirty water for long periods of time and drinking water directly from the river.

> "... because we men are always in water both day and night, pushing the boat to the river will need you to enter and sometimes it is dirty, thick with water weeds and that's why it's [bilharzia] more in men. For men, their life is more dependent on water where they go fishing both day and night, and that's why they are more likely to be infected than women. (Pakwach FGD Male, 46–65 years)

> "In some places like *Koppio* river, men normally go to very dirty points to place their hooks and can stay in water for some time. You may find that where you are standing has a lot of water snails making it easy for the worms to get into your body... There is this type of fishing, where you have to use the hooks but you have to go down into the water to get these big water snails to be used as bait.... Whenever they are fishing, they just drink this water direct. There is a channel where this water flows, it contains all sorts of wrong things that flow into the river. When you are removing these, you have to get them." (Pakwach FGD Male, 18–45 years)

In addition to actual fishing, men were responsible for catching the big snails (*koppa*) and worms required for baiting the fish, which exposed them as illustrated below:

> " ... to add on that, what makes us and other young men get this bilharzia is because of the activity we do in the water. We have some fishermen who do deep diving for about one to two hours underwater to mine or collect big snails [*koppa*] used for fishing as fishing bait. Under water. He keeps diving to collect the snails and comes out to pour them on the boat and dives again, and comes with snails placed between the neck and the shoulders and others stuck on his chest. That is what we have been doing for many years. We do that to feed our families. So, looking at the delay under water and the collection of these water snails

makes these worms [bilharzia] to enter in their body. It is different from snail mining. This is deep diving. And that is why most of them always have swollen/distended abdomens, and when you look at their skins, it is always dry and whitish. Some are dead. So, they swim back with snails packed from down upward. Oh, this system! We have come from far. (Pakwach FGD Male, 46–65 years)

". . . Men tend to look for various ways of survival more than women. One way is looking for life from water. Like digging of earth worms from mud used as fish bait. Those are earthworms used for fishing as bait. They first dig to get the earthworms from the mud. We have to enter in very dirty water, and that's where these worms tend to hide. Digging of earth worms. . .. They dig mud looking for these earthworms, which are later put on hooks as bait to get the fish." (Pakwach FGD Male, 18–45 years)

Furthermore, men also supplemented their fishing activities by mining small snails for the chicken industry:

"And another one is, there are some groups of wise people who came looking for these small types of snails for making chicken feeds. This has made us also get exposed because you have to enter deep in water, step on them and start scooping these snails which are the very carriers of bilharzia. It's still being carried out in Panyimur even now, but here we have stopped the activity. But people still go there to mine from Panyimur. They do it secretly. They value getting money". (Pakwach FGD Male, 46–65 years)

Women's fishing activities differed from those of men but were not necessarily safer. First, women got exposed from mining small snails and catching silverfish, both of which required them to get into deeper and dirtier water, sometimes for longer periods of time:

". . . women from here are involved in snail mining where they enter in deeper dirty water to scoop these snails with basin. This exposes them to get bilharzia" (Panyimur Female FGD 45–65)

"Sometimes back I used to scoop silverfish from the lake and would stay in the water for a long time, and would sometimes use bare hands to remove water snails away from the basin. Such activities would lead to infection by these worms." (Panyimur Female FGD 45–65)

However, women's most distinct role in the fishing industry was fish processing and trading as is illustrated below:

". . .The women are always interacting with the processing of fish and fetching water at landing sites while the men are always fishing twenty-four hours. Men bring the fish from the lake and pour them on the dry land, and the women start processing it by removing the intestines, scales, cleaning them in the lake, and taking them to the market. There is no way you can avoid women from using water in twenty-four hours; they always use water to do many things." (Panyimur Male FGD 46–65 years)

"Women do their businesses like buying fish, washing the fish, preserving fish, taking their time along the riverbank, removing the scales. That one also leads to high risk. Along the river bank we depend on the fish business, whereby fish business depends on struggle where you have to enter in water to get fish. Another thing is that we have a fishing mode called beach scening "korokota" which requires them to stand in water as they pull out the

net because it's the easiest fishing mode they can do, hence getting infected." (Panyimur Male FGD 18–45 years)

From the above, women were largely involved in fish processing and sometimes beach scening/*korokota*, which required women to get in contact with the water, buying fish and using lake water to wash the fish, thereby exposing them to infection. Apart from fishing, respondents mentioned other activities, particularly farming, which exposed men and women differently to infection:

"We men, sometimes we go and dig near the water bodies where these water snails are, to plant greens." (Pakwach FGD Male, 18–45 years)

"To add on that even digging at the banks is done by men where they may work on that piece of land for even one week, putting them at a greater risk of getting the disease. So there is farming, and some spots are only for the men to go." (Panyimur Male FGD, 46–65 years).

". . .I have found out that women can also get bilharzia. . . Yeah, we women can get it in many ways, more than men because the work we do using our hands are many like going to the farm to weed and may find somebody has openly defecated. The stool could be having bilharzia, you step on it or handle it unknowingly and get it, while most men get it from the river. We, women, have a lot of work that we do with our hands". (Pakwach Female FGD 46–65)

From the above, farming is portrayed as an activity which both women and men were involved in and which potentially exposed them to infection. While men farmed on the river banks where snails were, women farmed where people were likely to have defecated. Although there is hardly literature which cites touching or stepping in feaces as a potential source, the older Pakwach women's quotation above points to open defecation which is a threat when feaces are washed into the river by rain water.

### c) Gender differences in access to schistosomiasis treatment

In addition to exposure, gender differences were also reported with regard to treatment seeking behaviour of men and women. Treatment seeking behaviour refers to the different pathways people undertake to seek care if and when they feel ill. It is the active pursuit of good health, comprising a series of actions that an individual, their family member, or social network deploy to deal with ill health or disorder [25]. There are many factors that influence whether people will seek treatment, when and from who [26]? Some of these factors are patient related (including gender, education, religion, marital status, access to information, income, transport), while others are provider related (including the distance to the facility, availability of medicines, conduct of health care workers and many more). Gender differences have been documented, showing differences between men and women with regard to when and from where they sought treatment [27].

Generally, respondents knew that schistosomiasis was treated using PZQ and knew who needed to take it including children above the age of five years, infected persons, women who were not pregnant, snail miners, and fishermen. However, cases of non-adherence to treatment or delayed seeking of treatment were reported, which respondents attributed to the side effects of PZQ, the requirement to take PZQ after eating yet food was not regularly available especially in schools, and the infrequent supply of PZQ, only being provided in MDA when the WHO donated it.

    

Key informant interviews revealed that the biggest provider challenge was availability of PZQ. Accordingly, despite the policy of MDA of PZQ by the government, the medicines were not routinely supplied:

"... the availability of this medicine, it may not be easily accessible. You find right now I do not remember when they last distributed this medicine and am sure there are already people who need it. If they can make it routine, like this vaccination thing, it would work better for our communities around and it would reduce the high rate of the complications which people are getting from schistosomiasis, such as vomiting of blood and others" (KII, Male Health Facility In charge).

From this quote, there was infrequent supply of PZQ despite government's policy of door to door administration, which caused further complication of the disease. Moreover, many did not know they had the disease until they developed distended stomachs, by which time the disease would have advanced. Government relied on the WHO for donations of PZQ. As such it was not readily available in the hospitals, which patients interpreted as discrimination by the health care providers. And even when available in shops, its high cost was prohibitive for most households. As such, most were unable to purchase PZQ tablets when needed until the government provided them in MDA programmes. Meanwhile, many people tended to self-medicate thereby delaying entry into care.

In addition to PZQ scarcity, key informants listed several patient related factors including side effects of PZQ, which arose from taking it on a hungry stomach:

"There is one thing we learnt from these patients... some fear to swallow the medication, they talk about the side effects that the medicine has high side effects. Whenever they swallow, sometimes they vomit. At times when you get to direct observation therapy, most especially for children, they swallow it on empty stomachs like these kids from school. So, you find that the kids may swallow it on an empty stomach even when it disturbs them. But others, you may give them, and they say they will swallow later, but they may not even be in a position to swallow." (KII, Male Health Facility In charge).

This quotation highlights patient related factors which affect adherence. For example, even when school distribution of PZQ was considered the best mode of drug distribution, hunger threatened adherence, raising questions of adherence among the non-targeted category of adults.

With regard to gender, disparities existed with regard to treatment-seeking behaviours of men and women. For example, it was reported that men rarely sought treatment because of their economic activities, while women had opportunities to be treated whenever they took sick children to hospital:

"About the behavior, I think the men usually come to the facility in the late stages. When they are sick, they come when the disease has worsened, in the late stages of the disease and it becomes very difficult to treat them in our setting. So, when it comes to access to treatment men come at the stage which becomes difficult to treat them in our setting. For the women, accessibility to health care is easy because they are responsible for the children and the children fall sick more often than adults. So, whenever they bring the children to the health facility they also end up getting access to the services, testing and being treated. ... With men, usually they base a lot on their economic activity. For example, he will see the time he will come into the facility to get tested and treated as money lost. So, he will only

come to spend time in the facility when he is really sick and cannot do anything. If he is just feeling a little sick, he will just buy painkillers from a pharmacy then keep doing what he is doing. They prioritise money over health. Women usually come with children to the facility or for antenatal care when they are pregnant and in that way, they also get chance to be tested for bilharzia" (KII, Female Health Facility In charge)

". . . Sometimes it is the nature of service delivery, . . . sometimes they have to wait at the health centre to get treatment but men are impatient, unlike the women who always wait for service at the health centre." (KII, Male Health Facility In charge)

". . . for me it's hard. Even if it has been put at every village, I will miss it because most time am here working and I do not have time to move and will always hear that medicines has been given to people." (Panyimur Female FGD, 18–45 years)

These quotes reveal that economic activities were a hindrance to seeking health care for especially men and a few women who worked. Men's role as fathers/husbands who had to provide for families undermined their ability to promptly seek care. Accordingly, most men saw the time spent waiting and the money charged at the facility to be a waste. Hence for men, it was a choice between seeking care and making money, a choice between being a responsible man/father and being a patient. Men only sought treatment when they were too weak to work, by which time their conditions would have worsened and could not be handled at the facility, requiring referral to higher level facilities, which cost more in transport and health care charges.

Women's role as mothers, child carers and domestic workers enhanced their opportunities to enter into care earlier than men, as they visited facilities severally to take sick children for treatment, getting the opportunity to be tested and treated promptly. In addition, being home enabled them to access PZQ whenever it was distributed during the MDA. The revelation of the last quote by a working woman who revealed having not time to test or receive MDA shows that as more women work, they too are likely to miss out or enter into care late.

In spite of getting into care earlier, women relied on their spouses for decisions relating to health. One reason cited for this was illiteracy, where it was reported that they depended on their spouses to understand and utilize the health information accessed from the facility as illustrated below:

. . .One way you look at, in my experience, what I have seen is that more women can access information but because of limitation in education, the application of the knowledge is limited. A few people are literate. More men are more literate than women, so when they [men] get information they can use it faster. Culturally women are put in the backyard, therefore they have limited access to information. They spend most time at home doing housework. She will have no time to move to the facility more often except when taking children. Most child-rearing is for women. Also, men especially prefer to send boys to schools where they get information. Women need men's backing for finance and support, e.g. transport to the facility if far". (KII, Male Health Facility In charge)

From the above, illiteracy undermined women's chances of appropriately using the information. Relying on men for interpretation of health information received was likely to delay onset of care and reinforce gender stereotypes against women.

## Discussion

This study sought to conduct an intersectional gender analysis of vulnerability and treatment of schistosomiasis, a NTD using reported speech about a fairly poor rural community. It set out to show how gender intersected with other social stratifiers to shape experiences of vulnerability to and treatment for schistosomiasis. While it experienced challenges of applying intersectional gender analysis in a context where men, women and children are thought about in broad categories, it has been able to demonstrate the complex web of schistosomiasis infection and control (see Fig 1).

According to Fig 1, an individual's exposure to schistosomiasis infection and treatment of the disease occurred within a context of traditional gender roles (at family and societal level), limited economic opportunities due to poverty and insufficient health care services provision, in which schistosomiasis was a neglected tropical disease. Hence, the activities individual undertook were shaped by these factors, which in turn shaped their vulnerability to disease, and the decision to seek care.

The community had an understanding of the transmission pathways of the diseases as including the parasites, its vectors (snails), medium of transmissions (coming into contact with contaminated water), human activities that led to either contamination of water bodies (open defecation and urinating in the water or in the bushes) or economic and domestic activities which rendered people vulnerable to disease (including fishing, processing, farming, mining, swimming, and washing) and environmental factors such as rainwater flow.

From the findings, schistosomiasis exposure and treatment were gendered. Gender relations assigned different roles to men and women, which in turn affected the activities they performed within and outside the household, which in turn determined pathways to exposure to infection and access to treatment. But while men were perceived to be at the "epicentre" of the epidemic, women were not spared the risk of infection. The gender roles of either category, or the River Nile's centrality as the source of livelihood exposed all to infection.

The general discourse around men portrayed them as fathers and husbands who had to provide for their families while that around women portrayed them as mothers who had to

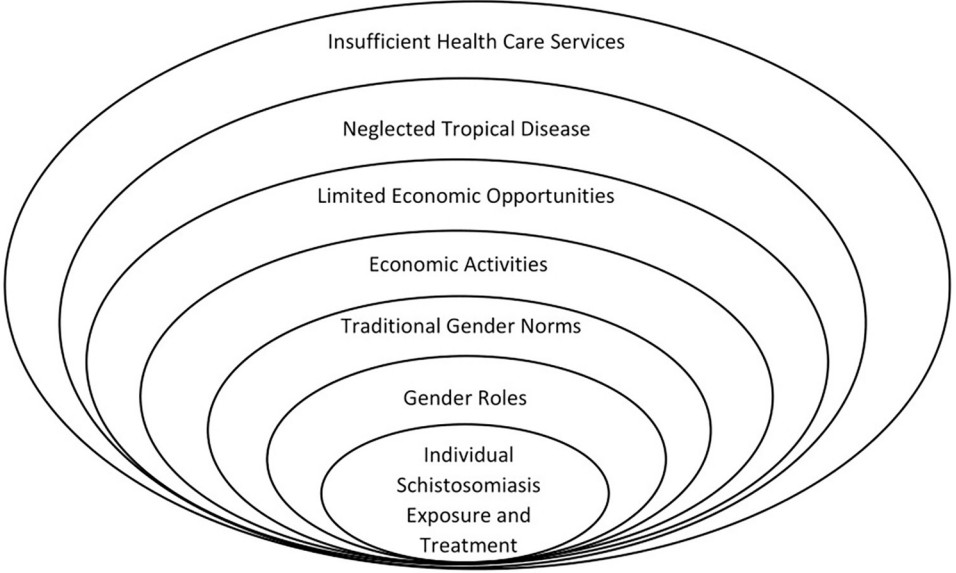

**Fig 1. Schistosomiasis Vulnerability and Treatment in Context.**

care for children and families. Unmarried or young men were obscured in this conceptualisation of masculinity, yet they undertook the most demanding and risky tasks in fishing such as deep diving for snails (*koppa*). The discourse around women or womanhood never distinguished between the married and unmarried, yet it was often reported that women with no working male household head or with a resident stay at home unemployed male had trouble surviving and had to take to fishing or working harder to provide for the home. Children were only mentioned in passing, particularly in reference to their vulnerability to infection from starting fishing early in life (for boys as young as 10 years) and fetching water for domestic use (for young girls). Children were also mentioned as vulnerable from playing in water and as targets for PZQ MDA in schools. Children's exclusion from gender discussions relating to schistosomiasis vulnerability and treatment obscure insights on how these really occur.

However, gender roles were not the only stratifiers. They were further conditioned by traditional norms, poverty and limited economic opportunities, as well as insufficient health care services, in which schistosomiasis remained under-prioritised. As with previous studies on other NTDs [18,28], MDA of PZQ for treating schistosomiasis targeted women in homes and children in selected schools and never targeted men at landing sites. MDA and or preventive chemotherapy has been adopted as the principle approach to providing essential medicines for NTDs in resource poor settings [29]. PZQ MDA is the main form of controlling schistosomiasis [13]. However, as with previous research which observed shortage of PZQ in Africa due to the high diseases burden [13], its infrequent supply threatened adherence and good treatment seeking behaviour. Hence, for men to miss out whenever the drugs were available because of work was to miss out on an opportunity to health. Yet, women who were considered to have opportunities to access care due to their gender roles were not in a better position considering that the medicines were scarce and that they relied on their male spouses to interpret the health information or for decision making relating to health. Unlike previous research, this was because of women's inability to read, and not the husband's gatekeeping as established elsewhere [17]. Moreover, women and children were targeted as an easy category without paying attention to the gender relations of exposure to, and treatment of the disease [18].

These findings also show the challenges of vertical health programmes. There is enough evidence about the limits of vertical health programmes and the need to integrate them with more community responsive strategies, including gender mainstreaming [18,28]. It is paradoxical that addressing diseases with so many gender and social norms affecting their prevalence are run in a vertical nature, to the exclusion of communities, especially women that would be central to their elimination. Such programmes end up targeting women and children to reduce morbidity without paying attention to gender power relations and leaving out the males who are responsible with household decision making regarding health. A gender mainstreamed strategy would consider the different roles of men, women and children and how these position them within the disease transmission, vulnerability and treatment cycle. For example Wenham and others [18] recommend gender mainstreaming in the fight against NTDs such as *zika*, while Geyer and others [28] recommend the integration of women in MDA against soil-transmitted helminths. All these point to the need to include communities, especially women in programming. Such a strategy would ensure working men and women are also reached for treatment as opposed to the current MDA strategy that only targeted those in schools and homes. It would also consider changes in gender roles, such as when women worked outside the home for income, are considered in programming instead of stereotyping women with domestic work. Finally, it would pay attention to gendered pathways through which children got infected and accessed treatment for schistosomiasis and other neglected tropical diseases.

Furthermore, there is need to prioritise NTDs and intensify their elimination. The PZQ stock outs in Africa where the schistosomiasis disease burden is high [13] shows the double marginalisation of those who suffer from it in Africa. Respondents in our study could be marginalised in four ways by having an NTD, being African, being located in a rural area of Uganda, and for women being female. Rhetoric has to be matched by resources, particularly with the routine provision of PZQ in Africa, where the disease is endemic.

In addition, there is need for more integrated health care services, including community sensitisation about the disease, its pathways and how it can be prevented and or treated. While in Pakwach there had been other interventions including the WASH programme, the persistence of schistosomiasis, calls for sustained government programming for the elimination of schistosomiasis and education of the people of its dangers. A wholistic outlook to health would consider treatment regimen, the need to target men at landing sites and public health campaigns around behaviour change (such as to combat openly defecating in water bodies and against drinking unboiled water).

Also, there was need to address poverty in the region. The region was characterised by historical marginalisation in terms of limited infrastructure, economic opportunities and services [30]. Located within this context, the gender relations observed produced an experience of disease which was dire. This calls for more development to provide alternate livelihood options to fishing or safer fishing and farming methods, which would protect them from further exposure to contaminated water bodies. Moreover, development would emphasise provision of appropriate domestic technologies to reduce on women's work burden, as well as improved motor boats to reduce men's stay on the water.

Lastly, academically, the study calls for continued understanding and expertise with regard to studying intersectional gender issues. This is especially so with regard to research methods such as FGDs and key informant interviews, which elicit community sanctioned views and not personal experiences. It is also pertinent in the context of not so obviously differentiated societies such as the research setting.

In conclusion, this study has highlighted the intersection of gender, poverty and limited health services in trapping people within the schistosomiasis vulnerability cycle. This complex web sustaining schistosomiasis and other NTDs includes the marginalisation of NTD's in global and national health systems programming, poverty in rural and underserved districts such as Pakwach, under-served rural populations with limited economic opportunities and health care services, as well as unequal gender relations. Pakwach being a poor and rural district had no means to guarantee safer livelihoods, water and sanitation amenities and uninterrupted PZQ drug supply. From the health systems perspective, the study confirms the limitations of vertical disease control programmes as well as project mode interventions. Apart from being temporal, such interventions have not adequately elicited the participation of the affected communities, whose change in norms or behaviour is crucial for eliminating the disease. This calls for gender sensitive pro-people health care systems, in which gender is mainstreamed in interventions for the prevention and treatment of schistosomiasis. This calls for more research in gendered pathways to infection and treatment of NTDs including schistosomiasis, in order to develop appropriate interventions to eliminate them.

## Supporting information

**S1 Data. Transcribed Interviews.**
(ZIP)

## Acknowledgments

The authors acknowledge Ministry of Health Vector Control Division and the management of Pakwach Health sub-district for the support that enabled this research.

## Author Contributions

**Conceptualization:** Sarah N. Ssali, Rosemary Morgan, Mariam Otmani del Barrio.

**Data curation:** Sarah N. Ssali, Christopher K. Opio.

**Formal analysis:** Sarah N. Ssali, Rosemary Morgan, Christopher K. Opio, Mariam Otmani del Barrio.

**Funding acquisition:** Sarah N. Ssali, Rosemary Morgan.

**Investigation:** Sarah N. Ssali, Christopher K. Opio.

**Methodology:** Sarah N. Ssali, Rosemary Morgan, Christopher K. Opio, Mariam Otmani del Barrio.

**Project administration:** Sarah N. Ssali, Salama Nakiranda.

**Supervision:** Sarah N. Ssali, Mariam Otmani del Barrio.

**Visualization:** Sarah N. Ssali.

**Writing – original draft:** Sarah N. Ssali.

**Writing – review & editing:** Sarah N. Ssali, Rosemary Morgan, Christopher K. Opio, Mariam Otmani del Barrio.

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
