## [Decision Letter · Decision Letter 0]

19 Oct 2022

Dear Associate Professor Ssali,

Thank you very much for submitting your manuscript "Gendered Lives, Gendered Vulnerabilities: An Intersectional Gender Analysis of Vulnerability to and Treatment of Schistosomiasis in West Nile Region, Uganda" for consideration at PLOS Neglected Tropical Diseases. As with all papers reviewed by the journal, your manuscript was reviewed by members of the editorial board and by several independent reviewers. In light of the reviews (below this email), we would like to invite the resubmission of a significantly-revised version that takes into account the reviewers' comments. 

We cannot make any decision about publication until we have seen the revised manuscript and your response to the reviewers' comments. Your revised manuscript is also likely to be sent to reviewers for further evaluation.

Sincerely,

Luc E. Coffeng, MD PhD

Guest Editor

Dileepa Ediriweera

Section Editor

Reviewer's Responses to Questions

**Key Review Criteria Required for Acceptance?**

**Methods**

-Are the objectives of the study clearly articulated with a clear testable hypothesis stated?

-Is the study design appropriate to address the stated objectives?

-Is the population clearly described and appropriate for the hypothesis being tested?

-Is the sample size sufficient to ensure adequate power to address the hypothesis being tested?

-Were correct statistical analysis used to support conclusions?

-Are there concerns about ethical or regulatory requirements being met?

Reviewer #1: Line 107- missing text after education. 

Line 115-118- it would be good to articulate the provision of PZQ in relation to mass drug administration campaigns as well as considering its availability within the open market (although the points about the open market are valid). 

Line 148-150- can you clarify what you mean by the long responses….

Description of gender, with sex then used as the purposive selection description- either change to sex in terms of participant identification or move toward terms men and women throughout the manuscript. 

Data analysis section- make a link here to the type of intersectional analysis applied and perhaps a few reflections on the process- as per comment above based on theoretical section.

Reviewer #2: Introduction

Line 53 – would be good to state the year of the prevalence of 25.6%. 

Methods:

Line 113 – what do you mean by customary beliefs limited the use of the water source?

Lines 113-115 – this is not clear to me about the fuel / wood. Can you please revise?

Lines 116 PZQ is on the WHO Essential Drugs list. See https://cdn.who.int/media/docs/default-source/essential-medicines/2021-eml-expert-committee/expert-reviews/i1_alb-meb-pzq_rev1.pdf?sfvrsn=91190dbf_5

Just because it is on the Essential Drugs list, doesn’t mean that it is available at the PHCs. Suggest rephrasing this sentence to state that although it is on the essential drugs list, it may not be available for routine treatment at the PHC outside the MDA. (If that is indeed what you are experiencing.)

In the study population section – you describe the study population as rural – but above authors say that the sites are urban and peri-urban. Please clarify. 

Data analysis – did you use any qualitative software?

**Results**

-Does the analysis presented match the analysis plan?

-Are the results clearly and completely presented?

-Are the figures (Tables, Images) of sufficient quality for clarity?

Reviewer #1: The results section is very interesting and there are some new and novel findings that are currently absent within the broader NTD literature presented. However, the balance of quotes to text and length of quotes makes the readability of the results very difficult to follow- could the authors select key quotes to include in the text and add others to a summary table or supplementary file- this would also prevent the reader from completing there own analysis of the research as is currently the case.

Reviewer #2: Results:

Would be nice to provide a summary of all data collected at the beginning of the section

Lines 230-231 – I am not sure that you can say young men are more culpable for transmission based on one quote. Rather that young men may have more freedom to move around, including defecation openly at night as there would be less social and government controls on their behaviours. 

Line 239-240 – do you need a reference for the definition of gender roles?

Line 247 – consider adding some specificity to the comment – on men’s role e.g., to ‘financially’ provide as opposed to only ‘provide’

Lines 304-355 – is there anyway to reduce the length or number of these quotes?

Lines 361 – lifting their clothes in the water will not necessarily expose them more to schistosomiasis infection. Once they have entered the water, they are exposed. 

Line 363 – authors have assumed that men go into the water 1x week, yet in the next section we hear from the men themselves that they are frequently in contact with the water. Please consider revising this statement. 

Line 453 – where is the evidence that men / boys are exposed earlier than girls? Girls will be washed in the river, will fetch water at early ages as well. This statement does not match with the evidence presented in the paper. The community perception may be that men / boys are more exposed due to the length of time they are in the water, or the fact that they don’t have a shirt on. But this is not scientifically valid.

Lines 583-5 – I am not sure that you can say that men are more likely to be infected than women. Both sexes have high exposure to contaminated water. 

Lines 619 – Table 1 - did the interviewers ask if men and women were equally at risk?

Line 633 – is a reference needed for the definition of treatment seeking behaviour?

Line 663 – PZQ is on the essential drugs list as noted above. The government however may choose not to purchase it for use in the PHCs. 

Lines 658-669 – are these comments directly related to the qualitative research?

Lines 713-725 – how is PZQ distributed? To the community or only to school aged children?

lines 842. Should be neglected tropical diseases which is more commonly used than neglected communicable diseases

**Conclusions**

-Are the conclusions supported by the data presented?

-Are the limitations of analysis clearly described?

-Do the authors discuss how these data can be helpful to advance our understanding of the topic under study?

-Is public health relevance addressed?

Reviewer #1: The limitations of the analysis are presented, however the situating of the findings within the broader NTD literature is absent- there is only one reference within the discussion and conclusions section. Would it be feasible for the authors to restructure the latter sections of the manuscript to streamline the results as described above and then have a separate discussion, recommendations and conclusion section. The discussion should link the findings to the broader NTD literature, in particular that which focuses on NTDs, gender, and intersectional analysis. For example see the special issue on gender and NTDs within this journal.

Reviewer #2: In general the policy implications reflect the data presented, but they could be written in a more succinct manner. Some specific comments below:

Policy implication:

a) Is not clear and needs to be written more succinctly. Is the focus on pregnant women? Fear of adverse events?

b) Schistosomiasis is not slated for eradication as per WHO recommendations. But for elimination. 

h) this may be an unrealistic recommendation 

i) this shift was demonstrated in a paper on LF MDA in Indonesia (Krentel and Wellings) 

The authors do not situate their findings within the broader literature on gender and infectious diseases or gender and NTDs. This is needed before moving into the policy implications and would further bolster these policy recommendations.

**Editorial and Data Presentation Modifications?**

Reviewer #1: (No Response)

Reviewer #2: Editing notes:

Please have consistency on the capitalisation of schistosomiasis. 

Line 102 – spelling – should be carved I think?

Line 107 – rest of sentence seems to be missing?

Line 137 – suggest another word – not ‘subjected’ – it sounds like a punishment 

Line 271-277 – reads awkwardly – please consider revising

Line 292 – health care worker (needs an -er)

LLine 740 – swam not swarm

**Summary and General Comments**

Reviewer #1: This is a really interesting paper and the analysis and application of gender theory is new and insightful. However, the limited link back to the NTD literature presents a slightly wasted opportunity in being able to make recommendations for action that could shape improvements in NTD care from a gendered perspective. I would encourage the authors to make these links more strongly as outlined in my comments above to maximise the impact of this work.

Further comments on the introductory section as follows: 

Line 40- the before world health organisation

Line 56-58- the list of NTDs affecting Uganda doesn’t seem exhaustive- perhaps the authors need to check against the full list of NTDs. For example, things like snakebite should be included. 

Line 60-63- the description of schistosomiasis control initiatives could be more streamlined- could the points about mass drug administration all sit together in the sentence prior to the alternative intervention approaches. 

Line 71-74- good text – perhaps better suited to methods section. 

The section on intersectional gender analysis is clear, however it would be good to include a couple of sentences on intersectional theory as well as the extensive description of gender analysis. For example, many would expect gender to be part of intersectional analysis implicitly- perhaps be firmer in articulating that gender is a key entry point and that you acknowledge the inherent need to consider gender in an intersectional analysis. Perhaps also describe more about your intersectional standpoint e.g. non-additive, intercategorical etc. See Christensen and Jensen 2012.

Reviewer #2: General comments - 

An interesting paper with useful results and exploration of how gender roles predispose men / women differently to infection and to treatment seeking. 

In some parts, authors should note sex not gender. 

Reduce the length of the paper – the quotes are long and repetitive. Is it possible to include some in an appendix or supplementary table?

PLOS authors have the option to publish the peer review history of their article (what does this mean?). If published, this will include your full peer review and any attached files.

Reviewer #1: No

Reviewer #2: No
---

## [Decision Letter · Decision Letter 1]

17 Aug 2023

Dear Associate Professor Ssali,

We are pleased to inform you that your manuscript 'Gendered Lives, Gendered Vulnerabilities: An Intersectional Gender Analysis of Exposure to and Treatment of Schistosomiasis in Pakwach District, Uganda' has been provisionally accepted for publication in PLOS Neglected Tropical Diseases.

Best regards,

Luc E. Coffeng, MD PhD

Guest Editor

Dileepa Ediriweera

Section Editor

Thank you for the revised manuscript and sincerest apologies for the delay in the review process. We tried to get the same reviewers to review your revised manuscript but only managed to get a recommendation from one of them.

To accept you paper for publication, please check your paper carefully one more time for accidental hard breaks (line 636? "furthermore" suggests a strong link with the preceding sentence/paragraph) and clusters of letters (e.g., lines 210-211).

Reviewer's Responses to Questions

**Key Review Criteria Required for Acceptance?**

**Methods**

-Are the objectives of the study clearly articulated with a clear testable hypothesis stated?

-Is the study design appropriate to address the stated objectives?

-Is the population clearly described and appropriate for the hypothesis being tested?

-Is the sample size sufficient to ensure adequate power to address the hypothesis being tested?

-Were correct statistical analysis used to support conclusions?

-Are there concerns about ethical or regulatory requirements being met?

Reviewer #1: No further comments on the methods- these are clear and previous comments actioned accordingly.

**Results**

-Does the analysis presented match the analysis plan?

-Are the results clearly and completely presented?

-Are the figures (Tables, Images) of sufficient quality for clarity?

Reviewer #1: The results have been amended and are now clearer to follow. At copy-editing i would suggest making these italicised to improve readability.

**Conclusions**

-Are the conclusions supported by the data presented?

-Are the limitations of analysis clearly described?

-Do the authors discuss how these data can be helpful to advance our understanding of the topic under study?

-Is public health relevance addressed?

Reviewer #1: The conclusions are supported by the data presented and the links to the literature within the discussion section are now clearer.

**Editorial and Data Presentation Modifications?**

Reviewer #1: (No Response)

**Summary and General Comments**

Reviewer #1: This is an important paper to progress discussions on gender and NTDs. I commend the authors on such an in-depth exploration of the topic. For future work, i would recommend moving away from the term intersectional gender analysis to either intersectional analysis or gender analysis. Intersectional analysis is inherently gendered and so it feels like saying the same thing twice. Additionally, for the future, more could be done to engage with the NTDs literature that is specifically feminist and focused on the issues here. That said, this is a good article and should move ahead to publication.

PLOS authors have the option to publish the peer review history of their article (what does this mean?). If published, this will include your full peer review and any attached files.

Reviewer #1: No

---

## [Editor Report · Acceptance letter]

3 Nov 2023

Dear Associate Professor Ssali,

We are delighted to inform you that your manuscript, "Gendered Lives, Gendered Vulnerabilities: An Intersectional Gender Analysis of Exposure to and Treatment of Schistosomiasis in Pakwach District, Uganda," has been formally accepted for publication in PLOS Neglected Tropical Diseases.

Best regards,

Shaden Kamhawi

co-Editor-in-Chief

Paul Brindley

co-Editor-in-Chief
